# Probiotic Effects against Virus Infections: New Weapons for an Old War

**DOI:** 10.3390/foods10010130

**Published:** 2021-01-09

**Authors:** Aroa Lopez-Santamarina, Alexandre Lamas, Alicia del Carmen Mondragón, Alejandra Cardelle-Cobas, Patricia Regal, José Antonio Rodriguez-Avila, José Manuel Miranda, Carlos Manuel Franco, Alberto Cepeda

**Affiliations:** 1Laboratorio de Higiene Inspección y Control de Alimentos, Departamento de Química Analítica, Nutrición y Bromatología, Universidad de Santiago de Compostela, 27002 Lugo, Spain; aroa.lopez.santamarina@usc.es (A.L.-S.); alexandre.lamas@usc.es (A.L.); aliciamondragon@yahoo.com (A.d.C.M.); alejandra.cardelle@usc.es (A.C.-C.); patricia.regal@usc.es (P.R.); carlos.franco@usc.es (C.M.F.); alberto.cepeda@usc.es (A.C); 2Área Académica de Química, Instituto de Ciencias Básicas e Ingeniería, Universidad Autónoma del Estado de Hidalgo, Carretera Pachuca-Tulancingo Km. 4.5, Pachuca 42076, Hidalgo, Mexico; josear@uaeh.edu.mx

**Keywords:** probiotic, gut microbiota, viral infection, viruses, influenza

## Abstract

This review aimed to gather the available literature investigating the effects of probiotics against the most common viral infections using in vitro trials in cell lines and in vivo clinical trials in both experimental animals and humans. Probiotics were employed to prevent and reduce symptoms of infections caused by common viruses, especially respiratory tract viruses, but also for viral digestive infections (such as rotavirus, coronavirus, or norovirus) and other viral infections (such as viruses that cause hepatitis, human papillomavirus, human immunodeficiency virus, and herpes simplex virus). Different probiotics have been studied to see their possible effect against the abovementioned viruses, among which different *Lactobacillus* species, *Bifidobacterium*, *Clostridium*, *Enterococcus*, and *Streptococcus* can be highlighted. In many cases, mixtures of various probiotic strains were used. Although the results obtained did not show similar results, in most cases, probiotic supplementation improved both barrier and biochemical immune responses, decreased susceptibility to viral infections, and enhanced the effects of concomitant vaccines. Works collected in this review show a beneficial effect of probiotics in the prevention and treatment of different viral infections. We found interesting results related to the prevention of viral infections, reduction of the duration of diseases, and decrease of symptoms.

## 1. Introduction

The mammalian intestine is a complex ecosystem, as it is a point of symbiosis between the host and approximately 10^14^ types of resident microorganisms, which have been acquired even before birth and continue to exist throughout life [1]. This community of microorganisms is often called gut microbiota (GM) [2,3]. The microbiome includes the microorganisms and their genetic material, which importantly contributes to host physiology by providing genetic elements that are not already present in the host genome [4]. Among GM microbes, some species can confer beneficial effects to the physiology and metabolism of the host. Among them, the Food and Agriculture Organization of the United Nations [5] defined the term probiotic as “live microorganisms, which when consumed in adequate amounts, confer a health effect on the host.” Probiotics are live bacteria that can be given as a supplement or in a food product that, if ingested in an adequate amount, can provide benefits to the host. Probiotics are composed mainly of lactic acid bacteria (LAB) and complex carbohydrate fermenters, which are part of the normal GM of humans and animals. Supplementation with probiotics can provide benefits to the host directly by preventing infection, or indirectly by enhancing the immune response of the host. There is scientific evidence that consumption of probiotics can play a role in increasing defense against external pathogens, thus maintaining the balance of the intestinal immune system [6,7].

Viral infectious diseases nowadays have a great impact on humankind. Viral infections produce variable morbidity and mortality, negatively affecting community health and causing wide economic losses [8]. The best example of this global threat may be the infectious disease caused by the new Severe Acute Respiratory Syndrome Coronavirus 2 (SARS-CoV-2), which during the 2019–2020 pandemic has infected millions of people worldwide [8]. Consequently, it is important to find alternative and safe ways to prevent viral infections and reduce the morbidity and mortality of viral infections. Even partially effective therapies for the treatment and prevention of viral infections can reduce the mortality, morbidity, and economic losses caused by these infections [9].

A large number of antiviral drugs that may be effective in certain infections are now available; however, the appearance of new viral strains through mutations is a major threat. Although combination therapies of drugs are effective against several viral diseases, it is always desirable to have additional approaches that may be used as preventives or supplemental therapies [10]. Among different strategies to prevent and reduce viral infections, adequate nutrition, including nutrients or food ingredients that enhance and potentiate immune response, are useful alternatives to reduce the number of infections and their severity. One nutritional strategy used in recent years to improve human immunity and reduce the ability to be infected is the intake of probiotics [11].

The main objective of this work was to provide a literature review of the effects of probiotic agents in the prevention and reduction of the severity of symptoms in viral infections, with special emphasis on the most recently published works.

## 2. Probiotics and the Immune System

According to the FAO consensus [5], criteria for probiotic bacteria include that the bacterial strain: (1) Must be able to survive in the gastrointestinal tract and proliferate in the intestine; (2) must benefit the host through growth or activity in the human body; (3) must be non-toxic and non-pathogenic; (4) must provide protection against pathogenic microorganisms that employ multiple mechanisms; and (5) must lack transferable antibiotic resistance. Bacterial strains of the same genus and species can have completely different effects on the host. Given this, the search for probiotic agents has evolved, and now it is even feasible to modify probiotics with increasingly complex functionalities by transferring mobile genetic elements [12]. Because probiotics can possess antiviral activity, they can be chosen as alternatives to or complementary antiviral therapies [13]. Although nowadays there is a significant demand for bioactive components such as probiotics in Western countries, consumers are reluctant to change their dietary habits [14]. Although in some cases probiotic products were developed by pharmaceutical industry, they have become increasingly popular among the public due to their inclusion in functional foods, generating wider acceptance among consumers [15]. This suggests that there is great potential for foods that are consumed regularly when they are converted to functional foods [14], that show better consumer acceptance than pharmacological presentations.

The most common genus of microbes used as probiotics are LAB, such as *Lactobacillus* [1], and primary oligosaccharide fermenters such as *Bifidobacterium* [16], although other genera, and even yeast, such as *Saccharomyces*, were proposed and used as probiotic agents [1]. One mechanism that probiotic strains, known as immunobiotics, can provide to human or animal health is modulating the mucosal and systemic immune systems [17,18], thus protecting against infectious diseases, including viral infections [13,19].

The consumption of probiotics has different beneficial effects on human health such as the production of antimicrobial and anti-adhesion substances against pathogens, thus facilitating the modulation of the immune system [20]. Probiotic bacteria can also inhibit the adhesion of the invading virus to the host-cell receptor by binding to it [13]. Moreover, probiotics can exert antiviral activity by direct probiotic-virus interaction, production of metabolites with antiviral inhibitory activity, or by stimulating the immune system of the host [13]. Probiotics may indirectly interfere with the virus by altering the state of cells, stimulating innate and adaptive immunity, or enhancing or suppressing associated molecular signaling pathways [21]. Probiotic bacteria can also have a protective effect against virus particles competing for adhesion to the cell surface. This has been tested in vitro and reported as a useful mechanism for cell protection in case of mucosal virus infections [22]. Another way of actuation of probiotics is the regulation of innate immunity using toll receptors and different signaling pathways, thus reducing inflammatory processes [23]. This regulation in innate immune system of the host can be kept increasing phagocytic activity, the activity of leukocytes (polymorphonuclears and monocytes), the expression of some receptors that are associated with phagocytosis, and the microbicidal function of neutrophils [13]. Different scientific studies showed that viruses, when they enter the body, activate an innate immune response where the inflammasomes are responsible for destroying the pathogens. The immune system detects pathogens in multiple ways, and there are two first-line of defense systems against viruses: The production of Type I interferons and interleukins (IL) IL-1β and IL-18 by inflammasomes. On the one hand, type I interferons promote an antiviral state in the infected host, and on the other hand, cytokines, including IL, induce inflammatory processes and modulate immune responses, producing antiviral effects [24]. For this reason, probiotic bacteria can regulate the activation of the inflammasome in organisms that have previously suffered inflammation due to viral infections. [24].

Another beneficial effect of probiotics is their capacity to help mature and activate the mucosal immune system by secreting metabolites such as organic acid, short chain fatty acids (SCFA), hydrogen peroxide, coagulation molecules, and bacteriocins, which are antimicrobial compounds [25,26]. These metabolites, particularly SCFAs, influence the gut epithelial and immune cells directly, enhancing the immune response. It was demonstrated that SCFAs reduced pattern recognition receptor (PRR) stimulation through activation of activated B-cell nuclear factor kappa-light-chain-enhancer and tumor necrosis factor-alpha (TNFα) [27]. In in vitro alveolar macrophages, it has been seen that the induction of low-level synthesis of nitric oxide can also influence the protective action of probiotic bacteria against viruses in respiratory cells [28]. Many studies have demonstrated that probiotics can increase the CD4+ lymphocytes count and regulate TNF-α, IL-6, IL-8, IL-10, and IL-12 [26].

## 3. Major Viruses Involved in Human Diseases

Nowadays, there are up to 200 species of viruses that can infect humans, and their number is increasing at a rate of 3–4 per year [29]. Consequently, most new human pathogens are viruses, and it is common for the virus to be of animal origin. A substantial proportion of mammalian viruses may cross the species barrier reaching humans. However, only a small number of these have human-to-human transmission and are therefore capable of causing human outbreaks. It will be practically inevitable that new human viruses will continue to emerge, and thus, an effective global surveillance system for new viruses is needed [29].

Although viruses can cause a wide variety of infections in different organs and systems of the human body, viruses were split, for this literature review, into three groups: Respiratory viruses, digestive viruses, and other viruses. Acute viral respiratory infections are among the leading causes of death worldwide, accounting for more than 4 million deaths per year [11]. In addition, these viruses are a leading cause of pediatric morbidity and mortality worldwide because of the immature immune system of the babies. On the other hand, the elderly are more susceptible to serious complications due to their weakened immune system [11].

These viruses spread easily between humans due to airborne transmission through aerosols, causing outbreaks that are very difficult to control [30]. Viral respiratory pathogens belong to various virus families, so RNA-containing viruses are more significant: *Picornaviridae*, *Orthomyxoviridae*, *Paramyxoviridae*, *Reoviridae*, *Coronaviridae*, and DNA-containing viruses, such as *Adenoviridae* and *Parvoviridae* [11,31]. Individuals can also be infected simultaneously with multiple viruses, and in some cases, multiple viruses have synergistic effects against host health [11,27].

Regarding viral gastrointestinal infections, different viruses, such as rotavirus (RV), norovirus (NV), or calicivirus (CV; including the agent Norwalk and astrovirus), can infect the human gastrointestinal tract and are responsible for many illnesses related to childhood diarrhea and gastroenteritis outbreaks worldwide [20]. These viruses have uncoated RNA, which makes them highly infectious and are transmitted in a fecal-oral manner. These infections are usually mild to moderate in severity and short in duration [32]. Rotavirus was traditionally the most common cause of severe dehydrating diarrhea in children, estimated to cause approximately 200,000 deaths in children under five years of age each year [19]. After the large-scale implementation of RV vaccines, NV is now the leading cause of severe diarrhea in children in developed countries and is also considered the most common cause of foodborne illness, nowadays associated with approximately 18% of gastroenteritis cases worldwide [32]. Astroviruses account for 2–9% of pediatric gastroenteritis cases worldwide [32]. There are enteric viruses that replicate in the gastrointestinal tract but are asymptomatic, such as reovirus or poliovirus, that can cause severe disease after spreading to peripheral tissues [32].

Another important group of viral infections affecting humans worldwide are liver infections. Among them, enterically-transmitted Hepatitis A (HAV), Hepatitis B virus (HBV), Hepatitis C virus (HCV), and Hepatitis E (HEV) viruses are causes of acute viral hepatitis in humans [33,34]. These viruses are transmitted through various routes, such as blood transfusions, sexual contact, and consumption of water or food contaminated by feces. HAV use the latter route for transmission; therefore, their outbreaks are more common in underdeveloped countries [35].

Skin viral infections include a large variety of viral agents, among which herpes zoster, which is caused by the reactivation in adults of the varicella-zoster virus, stands out. Herpes viruses can cause a primary infection, establish a latent infection in a specific set of cells in their host, and then reactivate when immunity weakens [36]. One of the most common members of the herpes virus family is cytomegalovirus (CMV), and more than 80% of primary infections occur in transplants. Herpes simplex virus (HSV) mainly affects the genital and perioral regions [36]. Another virus that causes human skin infections is the human papillomavirus (HPV), which presents several different types of warts, depending on the infected surface and its relative humidity, and pressure patterns [37]. Human papillomavirus represents a diverse group of viruses that primarily infect epithelial and mucosal tissues [38]. Polyomaviruses have been suspected as potential etiological agents in human skin cancer. These viruses infect epithelial tissues throughout the body, producing benign and malignant lesions, including common and genital warts [39].

Concerning viruses that can cause neurological infections, rabies virus (RABV) is one of the diseases that has been known since the beginning of civilization and has caused much fear. According to the World Health Organization [40], more than 59,000 and 21,476 deaths have been recorded worldwide and in Africa, respectively, due to this virus. Rabies is a disease of zoonotic origin caused by neurotropic viruses and is mostly spread by rabid animals, belongs to the genus lyssavirus, family *Rhabdoviridae* [41]. Other viral neurological infections are caused by arboviruses, which include several families of viruses that are transmitted by arthropod vectors. The arbovirus group includes *Flaviviridae*, *Togaviridae*, *Bunyaviridae*, and *Reoviridae* families that possess a high capacity to adapt rapidly to changing environmental and host conditions [42]. On the other hand, the most common cause of epidemic viral encephalitis in the United States today is West Nile virus infection. The incidence of this virus has increased significantly since 2008, especially in southern Europe. This and other mosquito-borne flaviviruses are considered endemic in Europe, such as Usutu virus [43].

There are also a large variety of hematological viruses, which cause infections with hemorrhagic fevers, that are RNA viruses encased in a lipid bilayer derived from the host′s cell membrane [44]. Viral hemorrhagic fevers are typically endemic in some regions, can cause large outbreaks, and have high mortality rates. They are characterized by an acute febrile syndrome with hemorrhages and affect both humans and animals [44,45].

## 4. Probiotic Usage against Respiratory Viruses

It has been demonstrated that the intestinal microbiota affects the health of the lungs due to the direct relationship between the microbiota and the lungs, known as the “gut-lung axis”. This axis is bidirectional, so endotoxins and microbial metabolites can affect the lung through the blood, and on the other hand, when inflammation takes place in the lung it can affect the intestinal bacteria [23]. Previous studies have found respiratory infections that are related with a change in the composition of the GM [23]. For example, mice with influenza viral infections in their respiratory tract have increased *Enterobacteriaceae* and reduced *Lactobacillus* and *Lactococcus* in their GM [46]. Secreted metabolites and immunomodulatory signals, such as secondary bile acids, secreted by commensal bacteria bind to their receptors in innate cells, such as macrophages, stimulating their metabolism and functions [23]. Additionally, it was demonstrated in murine models that removing some bacterial species from the GM by antibiotic treatments leads to an increased risk of influenza virus [46].

Disrupting the adhesion of the virus to mucosal cells could be beneficial to the host. Probiotic bacteria could bind directly to the virus, producing this disruption [25]. However, despite the abovementioned results about probiotic bacteria in the prevention and treatment of respiratory viral infections, nowadays there is no clear consensus about this matter, because on the one hand there are clinical trials that demonstrate the benefit of the use of probiotics in respiratory infections, but other clinical trials did not obtain any advantage after probiotic bacteria usage.

An important meta-analysis, with more than 8000 preterm infants included in several clinical trials, demonstrated that patients receiving enteral supplementation with probiotics showed a reduction in mortality caused by respiratory infections [47]. Viruses are especially important in respiratory tract infections because they cause more than 90% of upper respiratory tract infections [48]. The previous works regarding the effect of probiotic supplementation on viral respiratory infections is shown in Table 1. In most cases, the trials investigating this relationship were performed in mouse models [49,50,51,52,53,54,55,56,57,58,59,60,61,62,63], but some were also in clinical trials with children [64,65,66,67,68], adults volunteers [9,69,70,71,72,73,74,75,76,77], and the elderly [78,79]. The probiotic bacteria employed to prevent respiratory viral infections, in most cases, were *Lactobacillus* strains. However, other bacterial genera were also employed, such as *Clostridium* [69], *Bacillus* [69], *Enterococcus* [69], *Bifidobacterium* [25,65,67,68,77,80], *Streptococcus* [81], and *Propionibacterium* [67].

The lack of consensus on probiotic strains/gender may be due to differences in studies conducted and outcomes reported measures, the length of intervention, study populations used (children vs. adults), bacterial dose (10^6^–10^10^ CFU/day), or different matrices (milk, yogurt, capsules) used. Additionally, decreased immunity due to aging may partly explain the conflicting results in the elderly [9]. For instance, it was demonstrated that specific strains of lactobacilli could bind and inactivate flu-like respiratory virus in vitro [9]. Probiotic lactobacilli were reported to protect against respiratory tract infections by modifying innate and acquired host immune responses [50]. Additionally, a concrete lactobacilli strain (*L. plantarum* DK119) can prevent influenza A H1N1 and H3N2 infections and mortality in a mouse model, promoting innate host immunity to influenza infection by modulating alveolar macrophages and dendritic cells [51].

Genera other than *Lactobacillus* were employed as probiotic mixtures and not individually. In most cases, probiotics were applied against influenza virus infections, with the exceptions of Kumpu et al. [64], Luoto et al. [66], and Garaiova et al. [65], who investigated various viruses, Berggren et al. [70], who investigated probiotic effects against cold infections, and Guillemard et al. [73], who investigated effects against rhinopharyngitis. It is normal that the effects against influenza viruses have been the most investigated virus because the influenza virus produces a respiratory infection that is a major cause of morbidity and mortality worldwide, and new influenza subtypes become more dangerous to society. Studies have shown that oral and intranasal administration of LAB protects against this virus. In addition, LAB, such as certain species of *Lactobacillus* and *Bifidobacterium*, have been reported to modulate systemic and mucosal immune responses, mainly by improving mucosal Immunoglobulin A (IgA) production [49].

Additionally, due to the SARS-Cov2 pandemic, the effects of probiotics against this disease were recently investigated. One of the serious clinical manifestations of COVID-19, especially in the elderly and immunosuppressed, is pneumonia and the severity of acute respiratory distress syndrome [83], children suffer less severe symptoms [84]. Thus, it is reasonable to think that improving innate immunity could be useful in reducing the severity of this infection.

## 5. Probiotic Usage against Digestive Viruses

It is well known that some probiotics can prevent infections in the gastrointestinal tract and infections in other organs [48]. The main viruses that are involved in human diarrhea are rotavirus (RV), calicivirus (CaV; which includes norovirus (NV) and sapovirus (SAP)), enteric adenovirus, and astrovirus. Of these, RV and CaV are responsible for most cases of severe gastroenteritis [2]. RV is transmitted through contact between humans and responsible for one-third of the cases of severe diarrhea in children under five years old around the world [2]. Enteric viruses can be classified into two different mechanisms. One type multiplies in the intestinal epithelium, such as RV, which cause gastroenteritis, and the other type multiplies in the intestine, such as enteroviruses, but spreads to target organs, causing serious diseases [2].

In recent years, it has been shown that several LAB strains exert protection against infections produced by enteric viruses, increasing the production of specific antibodies and shortening the episodes of diarrhea [85,86]. Among its protective effects produced are mechanisms, such as increased immune defenses of the host and the production of antibiotic-like substances, IgA, and cytokine stimuli [86]. These probiotic strains have also been shown to increase mucosal secretions, improve intestinal motility, or enhance the productions of SCFA that act as protectors for the gut. There are also studies indicating that probiotic bacteria may have effects on the maturation of intestinal macrophages and dendritic cells and enhance cytokine production [86].

Table 2 shows different studies on the effect of probiotics and prebiotics against viruses that cause digestive diseases. These studies have been carried out both in vitro using cells lines [6,7,85,87,88] and in vivo using mouse models [80,89,90] and humans [91,92]. For these purposes, single *Lactobacillus* [88,90] or *Bifidobacterium* [80], but more frequently, mixtures of these two LAB with different bacterial genera, including also *Streptococcus* [87,92], and *Enterococcus* [7], were used. With respect to viral agents, the most common viral infections investigated were RV [6,7,79,89,92], transmissible gastroenteritis coronavirus (TGEV) [7,88], or murine NV [90].

As can be seen in Table 2, all the probiotic treatments demonstrated different degrees of protection against viral infections by different mechanisms, such as stimulating interferons, interleukins of Ig production, or accelerating reactive oxygen species (ROS) depletion in tissues. In some cases, probiotic strains also benefited the host against viral infection via genetic regulations modifying the expression of mRNA encoding viral polymerase [90] or genes involved in the inflammatory response [85].

## 6. Probiotics against Other Viruses

As was previously defined, there are a large variety of viruses that can cause other types of infections besides respiratory or digestive infections. Unfortunately, many of these cause very serious diseases that probiotics, and even in some cases the most modern medical technologies, cannot solve. However, for viruses that cause hepatitis, skin virus infections, human immunodeficiency virus (HIV), or HPV, probiotics could directly or indirectly, help reduce their symptoms or prevent infection

Table 3 shows the probiotic effects on the symptoms of other viral infections that were performed in mouse models [21,93], monkeys [22], or humans [26,94,95,96,97,98,99,100]. The trials performed investigated the effects of LAB, but also other less frequent species, such as *Enterococcus* [94], *Escherichia* [22], or even yeast, such as *Saccharomyces* [93]. With respect to viral infections, these probiotic agents were employed to improve the symptomatology of viral infections, such as HCV ([93], HPV [21,26,95,96], HIV [22,97,98], or herpes simplex virus (HSV) type I [22,93].

Regarding hepatitis infections, there are different studies that support the idea of including probiotics in routine therapy to fight fatigue, nausea, and low appetite, which are secondary effects of hepatitis C drug treatment [94]. An imbalance in intestinal microbiota has been seen in individuals with liver cirrhosis. Thus, for this reason, modifying the GM with probiotic supplements can decrease endotoxins and other compounds derived from bacteria, such as ethanol, phenol, and indoles, which are toxic and cause liver damage [94].

In HIV prevention, various observational studies have shown probiotic supplementation can prevent bacterial vaginosis caused by *Gardnerella vaginalis* and *Mycoplasma hominis*, a condition that facilitates the transmission of HIV [98]. Gori et al. [101] showed that in HIV-positive patients, there is usually a gastrointestinal deterrent at the beginning of the disease. This symptom is associated with, among other things, the GM disorders, which confirms a possible correlation between GM, gastrointestinal mucosal damage, and the immune system [102]. In HIV-positive people, there occurs an alteration of total microbial colonization as well as the microbiota composition in the oral cavity, and decreased CD4+ T cell counts have been associated with the presence of oral lesions [102]. Additionally, an intestinal disorder with pro-inflammatory effects has often been seen [97] with the increase in GM of either pro-inflammatory or potentially pathogenic bacterial populations, such as *Pseudomonas aeruginosa* and *Candida albicans*, whereas there is a reduction in beneficial bacterial counts such as *Bifidobacterium* and *Lactobacillus* [102]. An elegant review recently reported [103] investigated the effects of probiotics on inflammation markers in HIV patients, with no clear results due to the generally limited power of the studies included, that in most cases must be considered only exploratory.

Other authors reported that the prevailing HIV-associated dysbiosis across several cohorts seems to consist of enrichment of the phylum Proteobacteria including several subtaxa containing pathogenic bacteria, combined with a depletion of taxa within the bacterial families R*uminococcaea* and *Lachnospiracea*, known producers of SCFA [103]. A modification of the activity of the enzyme indoleamine 2,3-dioxygenase 1 induced by the interferon has also been described, which produces alterations in the tryptophan metabolism pathway [97]. These findings support the hypothesis that alterations in the gastrointestinal tract are a very important factor in the pathogenesis of HIV infection. In a study by Sheri et al. [97], it was shown that a mixture of probiotics, combined with IL-21, reduced the expression of several markers that activate CD4+ T cells. Because of this, it was suggested that probiotics act on the signaling of toll-like receptors (TLRs). Another important factor in the use of probiotics among HIV patients is that GM is crucial for the normal development of gut-associated lymphoid tissue (GALT), essential to ensure an appropriate immune activity [102].

However, the use of probiotics in severe immunocompromised or severely debilitated patients, such as HIV or recently transplanted patients [104]. HIV infection, at least during later stages, is associated with increased gut permeability and loss of local host immunity in GALT that may predispose to invasive infections. Thus, it was previously identified at least 11 single cases of patients, who after *Lactobacillus* supplementation suffered invasive *Lactobacillus* infections, causing symptoms such as bacteremia, pneumonia, or empyema, and even causing one documented death [104]. The risk of probiotics in immunocompromised people varies based on their contents, being *L. rhamnosus* the riskiest *Lactobacillus* species [105].

The idea of oral administration of probiotics against vaginal infections came from the knowledge that many urogenital infections arise from the entrance of a pathogen from the rectum to the perianal skin and then the vagina [106]. The mechanisms of action of probiotics at vaginal level include acidification of the mucosal surface, prevention of the adherence of pathogens, production of substances such as vitamins an immune modulators, and synergistic action with the host immune system [106]. Palma et al. [95] reported that HPV clearance was higher with the treatment of metronidazole and six months vaginal Lactobacillus implementation. On the other hand, Verhoeven et al. [96] previously failed to find any influence of probiotics on HPV clearance in a group of women with HPV-related low-grade squamous intraepithelial lesion using oral *Lactobacillus casei* Shirota (1 × 10^10^ CFU/day) for six months.

## 7. Conclusions

Recently published studies show the beneficial effects of using various probiotics to treat different diseases caused by viruses. Although they do not cure diseases, these probiotics are beneficial to patients because, in some cases, they improve the immune system and reduce the number of days of illness and the symptoms of the disease. Given the results obtained, probiotics can be an alternative for the prevention and treatment of many viral diseases that cause so many deaths around the world each year, or at least improve the quality of life of patients suffering from these diseases. Especially, there is great potential for probiotics consumed through functional foods, that seen to show better consumer acceptance than pharmacological presentations. There is a profound need for more in-depth studies into the benefits of the administration of probiotics in viral infections.

## Figures and Tables

**Table 1 foods-10-00130-t001:** Effects of probiotics against respiratory viruses.

Type of Study	Probiotics	Dosage and Time of Exposure	Viruses	Main Findings	Reference
In vivo using female BALB/c mice	140 different strains of lactic acid bacteria (LAB)	120 mg LAB/day for 28 days	Influenza A/X/31 (H3N2) virus	*Lactobacillus plantarum* AYA protects against respiratory influenza virus infection and decreased influenza lethality in mice	[49]
In vivo using 13 female BALB/c mice	Lyophilized *Lactobacillus rhamnosus* GG (LGG) and *Lactobacillus gasseri* TMC0356	10 mg of lyophilized LGG and *L. gasseri* for 19 days	Influenza virus A/PR/8/34 (H1N1)	The clinical symptom scores and pulmonary virus titers of mice administered oral LGG and *L. gasseri* were significantly ameliorated	[50]
In vivo using 96 elderly volunteers	Yogurt fermented with *Lactobacillus delbrueckii* ssp*. bulgaricus* OLL1073R-1 (1073R-1-yogurt)	100 g of 1073R-1-yogurt for 12 weeks	Influenza A virus subtype H3N2-bound	Consumption of fermented yogurt affected influenza A virus subtype H3N2-bound Immunoglobulin A (IgA) levels in saliva.	[78]
In vivo trial using female BALB/c mice	*L. plantarum* DK 119	Intragastric administration (200 µL of 10^8^–10^9^ colony count units (CFU) daily for 10 days) or intranasal (10^7^–10^9^ CFU/mouse)	H1N1 and H3N2 influenza viruses	*L. plantarum* protects against infection with H1N1 and H3N2 influenza viruses by enhancing the innate immunity of CD11c+ dendritic and macrophage cells and antiviral cytokines	[51]
In vivo using female BALB/c mice	*L. plantarum* 06CC2	20 mg/mouse, twice daily for 10 days	Influenza A/PR/8/34 (H1N1) virus	*L. plantarum* relieved influenza symptoms in mice in correlation with increased NK cell activity associated with increased production of interferon-α and Th1 cytokines through gut immunity and reduction of TNF-α in the early stage of infection	[52]
In vivo using 15 patients	*Clostridium butyricum* CBM588, *Bacillus subtilis* (unspecified strain), and *Enterococcus faecium* (unspecified strain)	Two tablets of probiotic compound were administered three times per day (~10^7^ CFU/tablet for CBM588 and 10^8^ CFU for *B. subtilis* and *E. faecium* enteric-coated capsules	Influenza virus H7N9	No beneficial effects have been seen in the administration of *C. butyricum* against H7N9 infection. Administration of *B. subtilis* and *E. faecium* improved the secondary infection.	[69]
In vivo using specific pathogen-free female BALB/c mice	*L. rhamnosus* M21 (KCTC 10965BP)	Oral administration of 0.3 mL of 1 × 10^9^ CFU/mL of *L. rhamnosus*	Influenza virus A/NWS/3 3 (H1N1)	*L. rhamnosus* increases the production of IgA and decreases the recruitment of inflammatory cells in the lungs, thus exhibiting anti-influenza activity by changing the host response to Th1	[53]
Clinical trial in in 272 subjects	*L. plantarum* HEAL 9 (DSM 15312) *and Lactobacillus paracasei* 8700:2 (DSM 13434)	Subjects were supplemented daily with either 10^9^ CFU of probiotics for 12 weeks	Common cold viruses	Oral intake of the strains *L. plantarum* and *L. paracasei* decreases the total symptom score and especially the pharyngeal symptoms of common cold infections	[70]
Clinical trial in 233 volunteers	*L. paracasei* N1115	Volunteers were given 100-mL bottles of yogurt, which contained living *L. paracasei* 3.6 x 10^9^ CFU, three bottles per day for 12 weeks	Viruses causing upper respiratory tract infections	The intake of yogurt containing *L. paracasei* could protect against the risk of acute upper respiratory tract infection in the mid-aged and elderly, might be that *L. paracasei* stimulated T-cell immunity	[71]
Clinical trial in 136 subjects	*L. paracasei, Lactobacillus casei* 431*, and Lactobacillus fermentum* PCC	All subjects received once-daily doses of probiotic drink (150 mL) that contained *L. paracasei* at 3 × 10^7^ CFU/mL, *L. casei* at 3 × 10^7^ CFU/mL, and *L. fermentum* at 3 × 10^6^ CFU/mL or placebo drink for 12 weeks	Viruses causing upper respiratory tract infections and influenza virus	Administration of these probiotics increased the levels of serum INF-g and IgA in the intestine. Reduced flu-like symptoms and the incidence of respiratory tract infection	[72]
In vivo using female BALB/c mice	*L. paracasei* CNCM I-1518	Mice were orally gavaged (200 µL) with *L. paracasei* (2 × 10^8^ CFU) daily for 7 days before infection	Influenza A/Scotland/20/74 (H3N2) virus	*L. paracasei* consumption seems to allow an early activation of proinflammatory cytokines (IL1α, IL-1β) and a massive recruitment of immune cells in the lungs after *L. paracasei* gavage and before influenza infection	[54]
Clinical trial in 69 children	*Lactobacillus acidophilus* CUL21 (NCIMB 30156*), L. acidophilus* CUL60 (NCIMB 30157)*, Bifidobacterium bifidum* CUL20 (NCIMB 30153)*, and Bifidobacterium animalis subsp. lactis* CUL34 (NCIMB 30172)	1.25 × 10^10^ CFU of probiotics plus 50 mg vitamin C or a placebo daily for 6 months	Viruses causing upper respiratory tract infections	Reduced incidence rate of respiratory tract infection symptoms in the probiotic group.	[65]
Clinical trial in 1000 volunteers	*Lactobacillus casei* DN-114 001	200 g/day for 3 months	Respiratory common infectious diseases	Reduced the risk of common infections in stressed individuals such as shift workers	[81]
Clinical trial in 94 preterm infants	*L. rhamnosus* GG ATCC 53103	1 × 10^9^ CFU/day for 1 to 30 days and 2 × 10^9^ CFU/day for 31 to 60 days	Adenovirus, coronavirus (229E/NL63 and OC43/HKU1), influenza A and B, Human metapneumovirus, parainfluenza 1, 2, and 3, RSV A and B, rhinovirus, Human enterovirus and bocavirus	The incidence of respiratory tract infections was lower in the probiotic group. The incidence of rhinovirus was significantly lower in the probiotic group. Incidence of rhinovirus-induced episodes tended to be lower in the prebiotic but not in the probiotic group	[66]
Clinical trial in 629 otitis-prone children	*L. rhamnosus* GG*, L. rhamnosus* Lc705, *Bifidobacterium breve* 99*,* and *Propionibacterium freudenreichii* JS	8-9 × 10^9^ CFU/day for 6 months	Human bocavirus 1-4 and rhinovirus/enterovirus	Lower number of human bocavirus 1 positive sample during the study, but no effect on rhinovirus/enterovirus occurrence	[67]
Clinical trial in 210 children	*B. animalis* subsp. *lactis* (BB-12)	10^9^ CFU/day for 3 months	Respiratory common infectious diseases	This study shows that *B. animalis subsp. lactis* has no effect on the prevention of respiratory tract infection in children. There was no significant difference in the number of people infected or in the duration of infection in the intervention group and the placebo group	[68]
Clinical trial in 97 daycare children	*L. rhamnosus GG*	10^8^ CFU/day for 28 weeks	Human bocavirus 1-4, rhinovirus/enterovirus, RSV, adenovirus, influenza A, and PIV 1-2	Respiratory symptoms decreased in children per month, but there was no effect on the occurrence of respiratory viruses	[82]
Clinical trial in 192 adults	*L. rhamnosus* GG *+ B. lactis* BB-12	5 × 10^9^ CFU of GG and 2 × 10^9^ of BB-12 CFU/day for 3 to 6 months	Human bocavirus, rhinovirus/enterovirus, RSV A and B, adenovirus, coronavirus (229E/NL63 and OC43/HKU1), influenza A and B virus, human metapneumovirus, and PIV 1-4.	Lower occurrence of rhinovirus/enterovirus after 3 months, but no significant effect on the occurrence of common respiratory viruses	[25]
Clinical trial in 209 adults	*L. plantarum* DR7	9 log CFU/day for 12 weeks	Viruses causing upper respiratory tract infections	Reducing plasma peroxidation and oxidative stress levels	[74]
Two clinical trials in 86 and 222 elderly volunteers	*L. casei* DN 114 001	Dairy drink (Actimel^®^) for 7 and 13 weeks	Influenza A (H1N1 and H3N2) and B	Daily consumption of this product resulted in increased specific antibody responses to influenza virus vaccination in persons over 70 years of age	[79]
In vivo using BALB/c mice (number not specified)	*L. rhamnosus* (unespecified strain)	Sublingually administered at 10^8^, 10^7^, and 10^6^ CFU/mouse for 3, 6, 10, 13, and 16 days	Influenza A/NWS/33 (H1N1)	Sublingual administration of *L. rhamnosus* increases the production of IgA in the secretion of the mucosa and the activity of T cells and natural killer cells, providing protection against flu virus	[55]
Clinical trial in 42 healthy adults	*L. rhamnosus* GG	Capsules containing 1 × 10^10^ CFU twice daily for 28 days	Influenza A (H1N1 and H3N2) and B	On day 28, a significant increase in seroprotection in the LGG group for the H3N2 vaccine strain was found	[75]
In vivo using BALB/c mice (number not specified)	*L. rhamnosus* GG (ATCC 53103*)*	Intranasally administered at 20 µL of LGG solution/day for three days	Influenza A/PR/8/34 (PR8, H1N1)	Intranasal administration of LGG enhances respiratory cell-mediated immune responses by following the activation of natural killer cells in the lungs, thus protecting the host from IFV infection	[56]
In vivo using 40 BALB/c mice	*Lactobacillus pentosus* strain b240	Oral administration of non-viable heat-killed b40 diluted at doses of 0.4, 2, or 10 mg/mouse/day for 22 days.	Influenza A/PR8/34 (H1N1)	Orally administered *L. pentosus* reduces influenza virus infectious titers in the lungs of influenza virus-infected mice	[57]
In vivo using BALB/c mice (5–6 per group)	*L. rhamnosus* CRL1505	Two consecutive days of 10^8^ CFU/mouse/day inoculated via nostrils using live and heat-killed *L. rhamnsosus*	Influenza A/PR/8/34 (H1N1)	Both viable and non-viable *L. rhamnsosus* reduced lung injury and viral load, protecting infected mice	[58]
In vivo using BALB/c mice (number not specified)	*L. pentosus* S-PT84	Intranasal administration of 20 µL of *L. pentosus* at a concentration of 0, 1, or 10 mg/mL once daily for 3 consecutive days	Influenza A/PR/8/34 (PR8, H1N1)	Intranasal administration of *L. pentosus* protected against flu virus infection by enhancing Th immunity, induction of INF-α and natural killer activity	[59]
In vivo using BALB/c mice (number not specified)	*Bifidobacterium longum* MM-2	Orally administered of 2 × 10^9^ CFU/day for 17 days from 14 days before 2 days after IFV infection	Influenza A/PR/8/1934 (PR8, H1N1)	Oral administration of *B. longum* stimulates immunity by increasing the activity of natural killer cells in the lungs and spleen, resulting in muffled viral proliferation. This probiotic suppresses inflammation in the lower respiratory tract, reduces symptoms, and improves the survival rate of IFV-infected mice	[60]
In vivo using 60 BALB/c mice	*Lactobacillus brevis* JCM 17312	1 × 10^9^ CFU/day for 14 days	Influenza A/PR/8/34 (H1N1)	*L. brevis* increases the production capacity of INF-α and the increase of the production of specific IgA of the human immunodeficiency virus, which can improve the symptomatology of this infection	[61]
Clinical trial in 50 volunteers	*L. fermentun CECT5716*	Oral daily dose of 1 × 10^10^ CFU 2 weeks before vaccination and 2 weeks after vaccination	Influenza A (H1N1 and H3N2)	In the probiotic group there was an increase in the production of natural killer cells, two weeks after vaccination. In addition, the antigen-specific IgA was also increased. The incidence of influenza-like illness was lower in this group 5 months after vaccination	[76]
Clinical trial in 211 subjects	*B. animalis* ssp*. lactis* BB-12(DSM15954), *L. paracasei* ssp*. paracasei, L. casei* 431 (ATCC 55544)	The probiotic products contained a minimum of 1 × 10^9^ CFU/day for 6 weeks	Influenza A virus	Both probiotic groups increased specific IgG and mean fold for vaccine specific secretory IgA in saliva	[77]
In vivo using C57BL/6N mice (number not specified)	*L. gasseri* SBT2055	Orally administered of *L. gasseri* at 1 × 10^8^ or 1.6 × 10^9^ CFU/mouse/day for 21 days	Influenza A virus (PR8)	Oral administration of *L. gasseri* improved the survival rates and the titer of the virus in the lungs, thus making the mice stronger against a viral infection	[62]
In vivo using BALB/c mice (number not specified)	*L. pentosus* b240	Orally administered heat-killed *L. pentosus* every day at a dose of 10 mg/mouse (10^10^) for 5 weeks	Influenza A (H1N1)	Expression of antiviral genes in rodent lungs can be regulated by administration of *L. pentosus*	[63]

**Table 2 foods-10-00130-t002:** Effects of probiotics against digestive viruses.

Type of Study	Probiotics	Dosage and Time of Exposure	Viruses	Main Findings	Reference
In vitro using a bovine intestinal epithelial cell line originally derived from fetal bovine intestinal epitheliocytes	*Lactobacillus gasseri* TMC0356, *Lactobacillus rhamnosus* (LGG), *L. rhamnosus* LA-2, *Lactobacillus casei* TMC0409, *Streptococcus thermophilus* TMC1543, *Bifidobacterium bifidum* 2-2, and *B. bifidum* 3-9	Lactobacilli or bifidobacteria (5 × 10^7^ cells/mL) for 24 or 48 h	Enteric common infectious diseases	Administration of *L. rhamnosus* induces the activation of TLR3, and there is an increase in the production of IFN-β by bovine intestinal epithelial cells, which may have beneficial effects on the protection against enteric viruses in vivo	[87]
In vitro using intestinal and monocyte/macrophage-derived cell lines (human, pig, goat)	*L. rhamnosus* (LGG), *L. casei, Enterococcus faecium* PCK38, *Lactobacillus fermentum* ACA-DC179, *Lactobacillus pentosus* PCA227, and *Lactobacillus plantarum* PCA236 and PCS22	10^8^ CFU/mL and incubated for 24-48 h	Rotavirus (RV) and transmissible gastroenteritis coronavirus (TGEV)	Administration of lactic acid bacteria (LAB) shows a protective effect against VR and TGEV. In the case of *L*. *casei*, Shirota has a high level of protection against TGEV by releasing highly reactive oxygen species (ROS) into the TLT cell line. *L. plantarum* PCA236 also stimulated the release of these reactive species	[7]
In vitro using a porcine intestinal epithelial cell line (PIE cells)	*Bifidobacterium longum* MCC1, *B**ifidobacterium infantis* MCC12, *Bifidobacterium breve* MCC16, *B*. *pseudolongum* MCC92, *Lactobacillus paracasei* MCC1375, *L*. *gasseri* MCC587, and *Lactococcus lactis* sub ssp. *lactis* MCC866	The cultured cells were incubated with different LAB strains at a density of 5 × 10^8^ cells/mL for 48 h.	RV	*B*. *infantis* MCC12 and *B*. *breve* MCC1274 increased the production of INF-β in PIE cells, in response to VR infection. They also increased the expression of CXCL10 and IL-6 genes, especially the *B*. *infantis*	[6]
In vitro using PIE cells	*L. rhamnosus* CRL1505 and *L. plantarum* CRL1506	Lactobacilli (5 × 10^8^ cells/mL) were added, and 48 h later effects were determined		Antiviral factors and cytokines/chemokines were increased in lactobacilli-treated PIE cells. The expression of the IL-15 and RAE1 genes that mediate poly (I:C) inflammatory damage was also reduced	[85]
In vivo using pregnant BALB/c mice	*B. bifidum* G9-1 (BBG9-1)	Orally administration of 3 × 10^7^ CFU of BBG9-1, respectively, once daily for 10 days from 2 days before to 7 days after RV infection	RV	The oral administration of *B. bifidum* induced mucosal protective factors, protecting against RV-induced lesions, and improving diarrhea. *B. bifidum* may be an effective method to control an RV epidemic for prophylactic and therapeutic purposes	[80]
In vivo using mice	Human-derived *Lactobacillus reuteri* DSM 17938 and ATCC PTA 6475	Diluted to a concentration of 2 × 10^9^ CFU/mL in PBS. Mice received gastric gavages (50 μL) of probiotics or vehicle daily from days 5 to 14 of life.	RV	A decrease in proinflammatory cytokine concentrations was seen, including the inflammatory protein of macrophages-1a and IL-1b, as well as an increase in the specific antibodies against rotavirus after the administration of the two probiotic strains*. L. reuteri* reduced diarrhea episodes	[89]
In vitro in ST cells	*L. plantarum* Probio-38 *and L. salivarius* Probio-37	10^8^ to 10^9^ CFU/mL	TGEV	Both strains survived in synthetic gastric juice and inhibited TGE coronavirus in vitro in ST cells	[88]
In vivo using 49 children	*L. casei* subsp. *casei* strain GG (LGG), *L. case*i subsp. *rhamnosus* (Lactophilus), or a combination of *S. thermophilus and L. delbrückii* subsp. *bulgaricus* (Yalacta^®^)	Twice daily for 5 days	RV	Administration of LGG increased the cells secreting specific IgA antibodies to rotavirus and in the convalescence stage. In addition, the duration of diarrhea was reduced in children	[92]
In vivo using 12 mice	*L. paracasei* ATCC 334	10^8^ CFU for 6 days	Murine norovirus (NV)	Intake of *L. paracasei* before the infection by murine NV, reduced the level of expression of the mRNA that encodes the viral polymerase	[90]
Clinical trial in 816 children	*L. rhamnosus* R0011 and *Lactobacillus helveticus* R0052	4 × 10^9^ CFU of *L. rhamnosus* and *L. helveticus* (95:5 ratio) twice daily for 5 days	Adenovirus, norovirus, and rotavirus	No beneficial effects associated with the administration of *L. rhamnosus* and *L. helveticus* have been observed; these probiotics do not reduce the severity of acute gastroenteritis or expedite the clearance of viruses in stool	[91]

**Table 3 foods-10-00130-t003:** Effects of probiotics against other viruses.

Type of Study	Probiotics	Dosage and Time of Exposure	Viruses	Main Findings	Reference
In vivo trial in 39 patients serologically positive for anti-hepatitis C virus (HCV) IgG antibodies	*Enterococcus faecalis* FK-23	900 mg of *E. faecalis* 3 times daily	HCV	*E. faecalis* decreased alanine transferase from 3 to 26 months of treatment while maintaining viral charge and other enzyme levels	[94]
Clinical trial in 180 women	*Lactobacillus rhamnosus* GR-1 and *Lactobacillus reuteri* RC-14 (50% each)	180 mg including 5.4 × 10^9^ CFU once a day until negative human papillomavirus (HPV) result	HPV	This probiotic may have decreased abnormal cervical smear rates, but it did not influence the genital burden of HPV	[26]
Clinical trial in 117 women	*L. rhamnosus* BMX 54 after a standard treatment of 500 mg metronidazole twice a day for 7 days	Vaginal tablets of 10^4^ CFU/tablet one each 3 days for 20 days and then once every 5 days for 2 months (short treatment), or once a week for 5 months (long treatment)	HPV	Probiotic implementation for 6 months favors the recreation of the vaginal balance, and therefore it can be useful to control the infection by the human papilloma virus	[95]
Clinical trial in 54 women	*Lactobacillus casei* Shirota	Daily consumption of a commercially available probiotic (Yakult^®^)	HPV	The likelihood of clearance of low-grade squamous intraepithelial lesion abnormalitieswas twice as high in the probiotic group	[96]
Clinical trial in 8 human immunodeficiency virus (HIV)-positive patients	Mix of *Lactobacillus plantarum* DSM24730, *Streptococcus thermophilus* DSM24731, *Bifidobacterium breve* DSM24732, *Lactobacillus paracasei* DSM24733, *Lactobacillus delbrueckii* subsp *bulgaricus* DSM24734, *Lactobacillus acidophilus* DSM 24735, *Bifidobacterium longum* DSM24736, and*Bifidobacterium infantis* DSM24737)	1.8 × 10^12^ CFU twice a day for 6 months	HIV	Administration of these probiotics decreases the level of tryptophan in plasma and increases the concentration of serotonin in the blood	[97]
In vitro trial in Vero African green monkey kidney cells	*L. rhamnosus* PTCC 1637 and *Escherichia coli* PTCC 25923	1 × 10^8^ CFU/mL	Herpes simplex virus-1 (HSV-1)	*L. rhamnosus* through various mechanisms, such as competition with the virus for adhesion to cells or increased viability of macrophages, induced antiviral effects against HSV-1	[22]
In vivo using 15 female C57BL/mice	*Bifidobacterium bifidum* (unespecified strain)	5 groups of 10, treatment groups were administrated either orally or intravenously with 100 μL *B. bifidum* (1 × 10^8^ CFU) 5 times at a 4-day interval for 20 days, including 2 times before and after tumor induction and one time on the same day of the challenge	HPV	Administration of this probiotic orally or intravenously, can modulate the immune system by stimulating secretion of INF-y and IL-12 in spleen cells and Th1 responses and prevent tumor growth	[21]
Clinical trial in 65 women with confirmed HIV infection	*L. rhamnsosus* GR-1 and *L. reuteri* RC-14	Daily capsules of freeze-dried probiotics with 2 × 10^9^ CFU and 400 mg of oral metronidazole twice daily for 10 days in women diagnosed with bacterial vaginosis	HIV	Administration of these probiotics can improve the quality of life of women with HIV-induced BV, but not cure it.	[98]
Clinical trial in 14 children	*L. plantarum* 299v	lyophilized powder in an oatmeal base in 5 g for 3 months	HIV	Probiotic bacteria can have protective effects against inflammation and activation of the gastrointestinal immune system by stabilizing the number of CD4+ T cells	[99]
Clinical trial in 39 subjects	*L. reuteri* MM2	1 × 10^10^ UFC/day for 21 days	HIV	No effects were detected in either safety or tolerance parameters	[100]
In vivo using male mice	*Saccharomyces boulardii* CNCM I-745	Oral gavage with either *S. boulardii* (10^7^ CFU/day) for 4 weeks	HSV-1	These probiotic increased levels of anti-inflammatory interleukins, decreased production of pro-inflammatory cytokines, and improved HSV-1	[93]

## Data Availability

The data presented in this study are available on request from the corresponding author. The data are not publicly available due to privacy concerns.

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
