# Peer review of "Probiotic Effects against Virus Infections: New Weapons for an Old War"

_foods, 2021, doi:10.3390/foods10010130_

Round 1
Reviewer 1 Report
The review entitled "Probiotic effects against virus infections: new weapons for an old war" is well written and conceived. I think that this review is a very interesting and actual topic. The authors highlighted the role of probiotics in counteracting different viral infections by describing several in vitro and in vivo literature results both in the text and tables.
This review is well-conceived and described.
Moreover, it is well written and I appreciated the division into three main virus types (respiratory, intestinal, and other viruses).
Author Response
Response: we sincerely value and appreciate constructive comments from the Reviewer
Reviewer 2 Report
In my opinion the subject of the manuscript is very interesting and well described. The manuscript needs corrections and additions.
The introduction is well edited and written, I have no objections to this part of the manuscript.
Below I present my comments and suggestions to the rest of the manuscript:
- Lines 81-82 - please note that Bifidobacterium are not strictly lactic acid bacteria
- Table 1, Table 2, and Table 3 - please note that in some examples only the species names of the tested bacteria are given, and the information about the strain used (and whether it was probiotic)
- In my opinion, the manuscript should emphasize the importance of this research review for food (taking into account the subject scope of the journal) and highlight those studies that deal with the antiviral effects of probiotics administered in the form of food (or feed for animals), but not pharmaceutical supplements
- I suggest supplementing the literature review with the following publications::
- doi: 3390/nu9060615
- doi: 10.1097/QCO.0000000000000612
- https://doi.org/10.2147/IDR.S210615
- Moreover, I believe that the safety of administering probiotics during viral infections should be discussed (i.e., see https://doi.org/10.1177/0956462415590725)
Author Response
With respect to the comments about “The introduction is well edited and written, I have no objections to this part of the manuscript”
Response: we sincerely value and appreciate constructive comments from the Reviewer
With respect to the comments about “Lines 81-82 - please note that Bifidobacterium are not strictly lactic acid bacteria”
Response: Thank you very much for your comment. In fact, Bifidobacterium are not strictly lactic acid bacteria. According to the suggestion from the Reviewer, in the revised version of the manuscript was changed to “primary oligosaccharide fermenters such as Bifidobacterium [16],”
Additionally, the following reference was added to the references list:
Luo, Y.; Xiao, Y.; Zhao, J.; Zhang, H.; Chen, W.; Zhai, Q. The role of mucin and oligosaccharides via cross-feding activities by Bifidobacterium: A review. Int J Biol Macromol 2020, https://doi.org/10.1016/j.ijbiomac.2020.11.087
With respect to the comments about “Table 1, Table 2, and Table 3 - please note that in some examples only the species names of the tested bacteria are given, and the information about the strain used (and whether it was probiotic)”
Response: Thank you very much for your comment. In fact in some cases of the works described in Tables 1-3 only the name of the species were stated and not the specific identification. In the revised version of the manuscript, it was added the specific nomenclature. However, some of the works included in the Review did not included the concrete code or nomenclature of the strain employed. In that cases, it was added the term “unspecified strain” after the name of the bacterial species.
With respect to the comments about “In my opinion, the manuscript should emphasize the importance of this research review for food (taking into account the subject scope of the journal) and highlight those studies that deal with the antiviral effects of probiotics administered in the form of food (or feed for animals), but not pharmaceutical supplements”
Response: Thank you very much for your comment. In order to emphasize the importance of consuming probiotics through functional foods, it was included the following paragraph in the revised version of the manuscript:
“Although nowadays there is a significant demand for bioactive components such as pro-biotics in Western countries, consumers are reluctant to change their dietary habits [14]. Although in some cases probiotic products were developed by pharmaceutical industry, they have become increasingly popular among the public due to their inclusion in functional foods, generating wider acceptance among consumers [15]. This suggests that there is great potential for foods that are consumed regularly when they are converted to functional foods [14], that show better consumer acceptance than pharmacological presentations.”
Additionally, it was included the following two reference sin the references list in order to reinforce the paragraph included:
Miranda, J.M.; Anton, X.; Redondo-Valbuena, C., Roca-Saavedra, P.; Rodriguez, J.A., Lamas, A.; Franco, C.M.; Cepeda, A. Egg and egg-derived foods: Effects on human health and use as functional foods. Nutrients 2015, 7, 706-729.
Díaz-Gutiérrez, L.; San Vicente, L.; Barrón, L.J.R.; Villarán, M.C.; Chávarri, M. Gamma-aminobutyric acid and probiotics: Multiple health benefits and their future in the global functional food and nutraceuticals market. J Funct Foods 2020, 103669.
With respect to the comments about “I suggest supplementing the literature review with the following publications”
Response: According to the suggestions from the Reviewer, the four articles suggested were included and discussed in the new version of the manuscript. In the Tables, we only included original experimental articles, and not reviews. Taking into account that all articles suggested by the Reviewer are reviews, they were included and discussed in the main text.
With respect to the comments about “Moreover, I believe that the safety of administering probiotics during viral infections should be discussed”
Response: According to the suggestions from the Reviewer, the following paragraph was included in the revised version of the manuscript:
“However, the use of probiotics in severe immunocompromised or severely debilitated patients such as HIV or recently transplanted patients [104]. HIV infection, at least during later stage, is associated with increased gut permeability and loss of local host immunity in GALT that may predispose to invasive infections. Thus, it was previously identified at least 11 single cases of patients, who after Lactobacillus supplementation suffered invasive Lactobacillus infections, causing symptoms such as bacteraemia, pneumonia or empyema, and even causing one documented death [104]. The risk of probiotics in immunocompromised people varies based on their contents, being L. rhamnosus the riskiest Lactobacillus species [105].”